# Inorganic Particles Contribute to the Compatibility of Polycarbonate/Polystyrene Polymer Blends

**DOI:** 10.3390/ma16041536

**Published:** 2023-02-12

**Authors:** Tetsuo Takayama

**Affiliations:** Graduate School of Organic Materials Science, Yamagata University, Yonezawa 992-8510, Japan; t-taka@yz.yamagata-u.ac.jp; Tel.: +81-236-26-3085

**Keywords:** mechanical properties, polycarbonate, polymer blends, injection molding, compatibilizer

## Abstract

Polycarbonate (PC), an engineering plastic, has excellent mechanical strength and toughness. Moreover, this transparent polymer material can be used in fields where materials require mechanical properties and transparency. Nevertheless, PC is known to have a high melt viscosity. Moreover, blending with polystyrene (PS), an inherently brittle material, has been used to adjust its melt viscosity. As a result, the PS makes PC/PS polymer blends more brittle than PC alone. As described herein, after attempting to achieve compatibility with inorganic particles, the results show that the dispersion of small amounts of inorganic clay and silica particles in PC/PS polymer blends maintained transparency while improving the impact strength to a level comparable to that of polycarbonate. Apparently, the inorganic particles promote the fine dispersion of PS. Moreover, the spherical morphology of the inorganic particles is more effective at compatibilizing the polymer blend because the inorganic particles can apply isotropic interaction forces.

## 1. Introduction

The glass transition temperature of polycarbonate (PC), approximately 150 °C, is higher than that of commodity plastics. As an engineering plastic known to have excellent mechanical strength and toughness, PC is a transparent polymer material that is used in fields and applications where transparency and certain mechanical properties are required. Unfortunately, PC has a high melt viscosity, making it difficult to process, which is important when it is regarded as a thermoplastic [1,2]. Furthermore, because PC is a polyester polymer, it is prone to hydrolysis. Therefore, even if a good product is obtained using PC with a small molecular weight, one can imagine that hydrolysis will progress over longer periods of use, leading to early embrittlement [3]. In other words, long-term durability is expected to be difficult to achieve. From the points presented above, one can infer that simply controlling the molecular weight is unlikely to resolve the inherent difficulties.

In addition to molecular weight control, polymer properties can be controlled by polymer blending, through which different polymers are combined. Because polymer combinations are fundamentally incompatible, polymer blends invariably form a phase-separated structure. In such a structure, an interface is formed inside the material. Therefore, if the refractive indices of the dispersed and continuous phases diverge and if the size of the dispersed phase is larger than the wavelength of light, then diffuse reflection of light occurs at the interface, thereby making the molded product opaque. From a mechanical point of view, product embrittlement occurs because a larger dispersed phase occurs as the specific surface area of the interface becomes smaller and because the yield condition of the molded product changes to interfacial debonding. This embrittlement markedly reduces the reliability of the molded product. Therefore, the dispersed phase must be dispersed to a size smaller than the wavelength of light to give the polymer blend transparency and high reliability. Compatibilization using additives is often used as a method to achieve the fine dispersion of the dispersed phase. Many of the additives used for this purpose are organic. Their effectiveness has been reported for various polymer blend compositions [4,5]. Organic additives are broadly classifiable into reactive and non-reactive systems, but their mechanisms are the same—they induce an action that controls the interfacial tension. For instance, Zolali et al. reported that the addition of poly(ether-b-amide) to a polylactic acid (PLA) polyamide 11 (PA11) polymer blend can control the dispersed phase and can form a co-continuous structure depending on the composition [4]. Nishino et al. reported that the addition of maleic anhydride-modified polystyrene to PC/ABS polymer blends improved the impact resistance [5].

Recently, the use of inorganic particles has attracted attention as an effective method for compatibilization [6,7,8,9,10]. Fang et al. reported that 1–5 phr of organically modified clay added to poly(ε-caprolactone) (PCL)/polyethylene oxide (PEO) polymer blends can finely disperse the dispersed phase [9]. Mederic et al. reported that 1 wt% of organically modified clay added to polyethylene (PE)/polyamide (PA) (PE)/polyamide (PA) polymer blends with about 1 wt% of organically modified clay effectively improves the mechanical properties [10]. A particle–matrix interface is created whenever inorganic particles are dispersed. Interfacial tension generated at the interface acts on the interface between the dispersed phase and the matrix, thereby reducing the interfacial tension and promoting the fine dispersion of the dispersed phase. This method is not constrained by the melt processing temperature. Therefore, its application to polymer blends is anticipated for processing temperatures at which applying organic systems is difficult. Compared to organic additives, which have been studied in the past, inorganic particles with nano-order particle sizes can be added in smaller amounts to achieve a fine dispersion of the dispersed phase, which is expected to improve the mechanical properties without compromising the optical properties.

Using the mechanism described above, this study was conducted to achieve the compatibilization of PC/PS polymer blends to adjust the melt viscosity, specifically with polystyrene (PS) blended with PC, and dispersed inorganic particles. Sako et al. reported that the transparency can be maintained while lowering the melt viscosity via blending with PC because the refractive index of PS is similar to that of PC [1]. Polymethyl methacrylate (PMMA) is a transparent thermoplastic similar to PS, but depending on its molecular weight, it may not be possible to produce transparent molded products when blended with PC [11,12]. Some attempts have been made to improve the mechanical properties of PMMA by increasing its reaction compatibility, but there is concern that its moldability may be inferior to that of PC due to its increased viscosity [13]. From the above points, it was considered that PS would be more effective in achieving the objectives of the study. In contrast, Kunori et al. reported that because PS is an inherently brittle material, PC/PS polymer blends are more embrittled than PC [14]. This study examined whether the embrittlement caused by PS blends can be ameliorated by the dispersion of inorganic fine clay or silica particles. In addition, to maintain the transparency of PC, the amount of dispersed inorganic particles should be extremely small. Therefore, in this study I developed a molding process to reduce the number of inorganic particles to 0.3 wt% or less. The addition of minute amounts of inorganic particles described here is a method that enables the compatibilization of polymer combinations that could not previously be expected to have synergistic effects. The application of this technology will lead to the creation of more multifunctional polymer molded products.

## 2. Materials and Methods

### 2.1. Materials

Polycarbonate pellets and polystyrene pellets were used as the raw materials. The melt flow rate of the PC was 10 g/10 min at 240 °C and 49.03 N. That of PS was 9.0 g/10 min under the same conditions. Clay (RXG7540; BYK GmbH, Wesel, Germany) and silica particles (Admanano YA010C; Admatechs Co. Ltd., Shanghai, China) were used as the inorganic particles. The clay particles before kneading were a mixture of fibrous and plate-like particles. The plate-like clay layer thickness was 1 nm. The surfaces of the 10-nm-diameter silica particles were treated with phenyl silane.

### 2.2. Melt-Blending

These raw materials were loaded into a twin-screw melt extruder manufactured in Japan (IMC0-00; Imoto machinery Co., Ltd., Kyoto, Japan) and were melt-kneaded to obtain polymer blend pellets. As the first step, PS and inorganic particles were melt-kneaded at a ratio of 90:10 wt% at 230 °C and 60 rpm to obtain masterbatch pellets. As the second step, the PC, PS, and masterbatch were adjusted to the specified mixing ratio; filled into the mixer; and melt-kneaded at 280 °C and 60 rpm to obtain PC/PS polymer blend pellets in which inorganic particles were dispersed. Table 1 presents the kneading ratios. Sako et al. reported that a 10 wt% blend of PS with PC can provide a sufficient viscosity reduction to improve the moldability [1]. In this paper, based on this report, I fixed the PC/PS ratio at 90/10 and focused on improving the mechanical properties of this blend.

### 2.3. Viscosity Measurements

Pellets obtained by melt kneading were filled into a melt flow indexer manufactured in Japan (G-01, Toyo Seiki Seisaku-sho, Ltd., Tokyo, Japan). Then, the melt viscosity measurements were conducted at 240 °C and 49.03 N to ascertain the melt flow rate (MFR).

### 2.4. Injection Molding

The pellets obtained by melt-mixing were filled into a small electric injection molding machine manufactured in Japan (C, Mobile0813; Shinko Sellbic Co., Ltd., Tokyo, Japan) and were injection-molded to produce test specimens. Figure 1 depicts the specimen shape. Specimens of two types were prepared: (a) dumbbell-shaped and (b) beam-shaped. Table 2 shows the injection molding conditions.

### 2.5. Tensile Tests

The dumbbell-shaped specimens were subjected to uniaxial tensile testing in accordance with ISO 527-1. The uniaxial tensile test was performed on a small universal mechanical testing machine manufactured in Japan (FSA-1KE-1000N-L, Imada Co., Ltd., Aichi, Japan). The loading speed was 10 mm/min. The distance between chucks was 22 mm. The load obtained in this test was designated as *P*. The nominal stress *σ* was obtained using Equation (1):(1)$$\sigma =\frac{P}{A}$$

Here, *A* represents the cross-sectional area of the parallel section of the specimen. The true strain *ε* was obtained using Equation (2), where *δ* denotes the obtained displacement:(2)$$\varepsilon =\ln\left(1+\frac{\delta }{L}\right)$$

The true strain at break was evaluated as the elongation at break. Five tests were conducted for each sample. The mean of the evaluated physical properties was used as the result for this report.

### 2.6. Determination of Elastic Properties

Three-point bending tests were performed on the resulting beam-shaped specimens in accordance with ISO 178. Three-point bending tests were performed on a tabletop tensile and compression testing machine manufactured in Japan (MCT-2150, A&D Co., Ltd., Tokyo, Japan). The loading speed was 10 mm/min. The span was 40 mm. The load obtained in this test was *P_f_*. The bending stress *σ_f_* was calculated using Equation (3):(3)$$\sigma_{f}=\frac{3P_{f}S}{2bh^{2}}$$

In the above equation, *S* stands for the span, *b* denotes the beam specimen width, and *h* expresses the thickness of the beam specimen. The bending strain *ε_f_* was calculated using Equation (4), with the deflection obtained in this test as *δ_f_*:(4)$$\varepsilon_{f}=\frac{6\delta_{f}h}{S^{2}}$$

From the *σ_f_*-*ε_f_* curve obtained from the three-point bending test, 2/3 of the value of the maximum point was obtained as the flexural yield initiation stress *σ_fy_*. The initial slope of the curve was determined as the flexural modulus *E_f_*.

The tensile deformation produced by the three-point bending test is regarded as pure tensile deformation. The yield initiation stress obtained in this case is determined by Equation (5):(5)$$\sigma_{fy}=\sigma_{y}\left(1+\upsilon \right)$$

Here, *υ* is Poisson’s ratio; *σ_y_* in the equation is the yield initiation stress obtained under uniaxial tensile loading. Assuming that the shear yielding of polymeric materials is attributable to intermolecular friction, the shear yield initiation stress can be obtained using Equation (6) [15]:(6)$$\tau_{y}=\alpha \left(T_{inj}-T_{test}\right)E\cos\theta$$

In the above equation, *α* represents the average coefficient of linear thermal expansion obtained in the range from the molding temperature to the test temperature, *T_inj_* stands for the injection molding temperature, and *T_test_* represents the test temperature. In addition, *θ* denotes the shear angle, which is calculated using Equation (7).
(7)$$\theta =\tan^{-1}2\upsilon$$

Because the yield condition of PS follows the maximum shear stress condition proposed by Tresca, *σ_y_* is defined according to Equation (8) [16]:(8)$$\sigma_{y}=2\tau_{y}$$

However, the yield condition of PC follows the shear strain energy condition proposed by Von Mises. In this case, *σ_y_* is expressed by Equation (9) [14]:(9)$$\sigma_{y}=\sqrt{3}\tau_{y}$$

Because the two equations above are the same except for the different coefficients, coefficient *B* is used in the following equations. Solving Poisson’s ratio from the equations above yields Equation (10) below:(10)$${\left\{\frac{{\left(1+\upsilon \right)}^{2}\left(1-2\upsilon \right)}{1-\upsilon }+\frac{6}{5}{\left(\frac{h}{S}\right)}^{2}\frac{{\left(1+\upsilon \right)}^{2}\left(1-2\upsilon \right)}{\upsilon }\right\}}^{-1}\cos\theta =\frac{\sigma_{fy}}{B\alpha \left(T_{inj}-T_{test}\right)E_{f}}$$

Because the right-hand side of this equation is already valued, Poisson’s ratio can be identified by establishing this equation. Furthermore, *E* is obtained using Equation (11).
(11)$$E=\left\{\frac{\left(1+\upsilon \right)\left(1-2\upsilon \right)}{1-\upsilon }+\frac{6}{5}{\left(\frac{h}{S}\right)}^{2}\frac{\left(1+\upsilon \right)\left(1-2\upsilon \right)}{\upsilon }\right\}E_{f}$$

Using the methods presented above, *υ* and *E* were found for each of the fabricated samples. Three-point bending tests were conducted five times for each sample. Then, the mean and standard deviation of the evaluated flexural strength and flexural modulus were used as the reported results.

### 2.7. Transmittance

Transmittance measurements were performed with a transmittance meter manufactured in Japan (TL-110V; Tokai Optical Co., Ltd., Aichi, Japan) using beam-shaped specimens obtained by injection molding. Measurements were taken in the range of 360 nm to 900 nm. The wavelength dependence of the transmittance was evaluated.

### 2.8. Charpy Impact Tests

Notched Charpy impact tests in accordance with ISO 179 were performed with beam-shaped specimens obtained by injection molding with notches machined in the width direction. This test was performed using a pendulum impact tester manufactured in Japan (Mys-tester Co., Ltd., Osaka, Japan). The span was 40 mm. The test speed was 2.91 m/s. The impact energy *U* was calculated from the obtained swing angle. The Charpy impact strength was calculated using Equation (12):(12)$$a_{iN}=\frac{U}{h\left(b-a\right)}$$

In the above equation, *a* denotes the notch depth. Five tests were conducted for each sample. Then, the mean and standard deviation of the evaluated physical properties were collected as the results reported herein.

### 2.9. Morphology Observation

The beam-shaped specimens obtained by injection molding were cut with a microtome until the core layer was reached. The specimen phase structure was observed using atomic force microscopy on a system manufactured in Canada (AFM, nGauge AFM, ICSPI) after cutting.

### 2.10. Differential Scanning Calorimetry (DSC)

Differential scanning calorimetry on a system manufactured in USA (DSC, DSCQ2000, TA instruments Co., Ltd., New Castle, DE, USA) was performed by cutting specimens from injection-molded beam samples. These measurements were carried out on PC and PC/PS, as well as the most improved compositions of PC/PS/clay and PC/PS/silica. Heat flow curves were obtained by increasing the temperature from 40 °C to 250 °C at 10 °C/min in a nitrogen atmosphere, and the intermediate glass transition temperatures of the PS and PC were determined from the obtained results.

### 2.11. Fourier Transform Infrared Spectroscopy

Fourier transform infrared spectroscopy (FT-IR) was performed on injection-molded beam samples. FT-IR was performed using a Fourier transform infrared spectrometer manufactured in Japan (Nicolet iS5, Thermo fisher Scientific K.K., Tokyo, Japan). The measurements were carried out based on the attenuated total reflectance (ATR). In the obtained absorption spectra, I focused on 695 cm^−1^, which indicates the out-of-plane CH broadening angle of the benzene ring of PS, and 1452 cm^−1^, which indicates the in-plane CH2 symmetry broadening angle. Through the study of the changes in the absorption of these two spectra, I investigated the effect of the interaction force generated by the inorganic particles on the dispersed phase of PS.

## 3. Results and Discussion

### 3.1. Morphology of PC/PS Polymer Blends

Figure 2 portrays the morphologies observed using AFM. The fine particle dispersion reduced the domain size until it became invisible under AFM examination. These findings indicate that the dispersed phase can be a small number of finely dispersed inorganic particles in PC/PS. The reason for this finding can be discussed in terms of the forces interacting between the dispersed phase and the inorganic particles.

Dispersing the inorganic particles creates a matrix phase–particle interface, where interfacial tension occurs. In this case, the inorganic particles were dispersed in the PC—the matrix phase. Therefore, the interface between the PC and the inorganic particles presumably created new interfacial tension, which increased the interfacial tension between the PS and the PC, leading to the fine dispersion of the PS.

Figure 3 shows the FT-IR results and the dependence of the inorganic particle addition on the absorbance at 695 cm^−1^ (*A*_695_) and 1452 cm^−1^ (*A*_1452_). Both absorbance values show a tendency to decrease with the addition of inorganic particles, with the clay showing a greater decrease. This suggests that clay has a stronger interaction force and strongly constrains the PS molecular chain. This may be due to the larger size of the primary particles of clay, which may result in a higher intermolecular force per particle.

### 3.2. PS Blending Effects on Viscosity and Transmittance

Figure 4 shows the MFR results. The MFR was increased by blending 10 wt% of PS with PC. It was decreased by dispersing a small number of inorganic particles into this system. Figure 5 presents the transmittance results. The transmittance at short wavelengths was decreased by blending 10 wt% of PS. Furthermore, even a slight clay dispersion, because of the light absorption property of clay, tended to decrease the transmittance. However, small amounts of silica did not affect the transmittance of short wavelengths, and actually increased the transmittance of long wavelengths.

### 3.3. Glass Transition Temperatures of PC/PS Polymer Blends

Table 3 shows *T*_g,PC_, the glass transition temperature of PC, and *T*_g,PS_, the glass transition temperature of PS, as obtained from the DSC measurements. The results for pure PC and PS are also shown in the same table. *T*_g,PC_ was shifted to the lower temperature side by blending 10 wt% PS, while *T*_g,PS_ was shifted to the higher temperature side. This is believed to be due to the negative pressure generated in PC during the cooling and solidification process by blending PS, which has a lower glass transition temperature than PC [17]. The *T*_g,PC_ was shifted to the high temperature side by adding an appropriate amount of inorganic particles. This suggests that the negative pressure exerted on PC is relieved by the dispersion of inorganic particles.

### 3.4. Mechanical Properties of PC/PS Polymer Blends

Figure 6 shows the nominal stress–true strain curves obtained from the tensile tests. The yield stress was decreased by blending 10 wt% of PS. Additionally, the true strain at break was increased. The results show that a small amount of fine particles decreased the strain.

The results presented above indicate that the small dispersion of inorganic particles contributes to the compatibilization of the polymer blend. That better compatibility in turn promotes shear deformation of the molded product, leading to toughening.

The uniaxial tensile test results showed that PS blended with PC tends to increase the elongation at break, indicating that the dispersion of inorganic particles in small amounts leads to a decrease in elongation at break. This apparent contradiction can be discussed in terms of the fracture criteria for the respective tests.

In the uniaxial tensile test, necking occurred in all compositions because of loading. The necking propagated and caused large deformation until strain hardening occurred, after which the material ruptured. In uniaxial tensile tests, the mechanism of change from interface debonding to shear yielding discussed in the notched impact test does not apply. It is more natural to consider that some property values change within the same deformation mechanism. Stresses generated in uniaxial tensile test are divisible into two main categories: extensional stress and expansive stress. The latter contributes to rupture, which occurs when this stress reaches a critical value. When porosity is defined as *f*_0_, the relation between the expansion stress *σ_v_* and expansion strain *ε_v_* is expressed as Equation (13) shown below [18]:(13)$$\sigma_{v}=\frac{2}{3}\frac{E}{\left(1+\upsilon \right)}\left(\frac{1-f_{0}}{f_{0}}\right)\varepsilon_{v}$$

For this equation, *f*_0_ is obtained from Equation (14):(14)$$f_{0}=\frac{1}{1+\frac{E}{4(1+\upsilon )\sigma_{y}}}$$

Equation (15) shows the expansion strain produced by the uniaxial tensile loading:(15)$$\varepsilon_{v}=\varepsilon \left(1-2\upsilon_{0}\right)+\varepsilon^{2}\left(\upsilon_{0}^{2}-2\upsilon_{0}\right)+\upsilon_{0}^{2}\varepsilon^{3}$$

Here, *υ*_0_ is defined according Equation (16):(16)$$\upsilon_{0}=\varepsilon^{-1}\left\{1-\sqrt{\frac{1}{1+\varepsilon }}\right\}-D$$

The equation above relies on the assumption that the material under plastic deformation is mostly incompressible fluid. The correction factor *D* for Poison’s ratio when the material is an incompressible fluid is introduced in the form of a difference as 0.01. The relation between the elongation at rupture and the expansion stress at rupture obtained from Equation (13) is shown in Figure 7. The expansion stress calculated using Equation (13) is positively correlated with the elongation at break. The y-intercept in this figure signifies the amount of plastic deformation in the absence of expansion stress—the strain required for necking to propagate. Based on the relation between the expansion stress and tensile stress obtained in the uniaxial tensile tests, the tensile stress at rupture, *σ_B_*, can be expressed as Equation (17).
(17)$$\sigma_{B}=3\sigma_{v}$$

The fracture stress of PC obtained with this equation was approximately 165 MPa. Although this stress is slightly higher than the reported breaking stress, this method reproduces the approximate value [19]. Figure 8 shows the relation between the fracture stress and the inorganic particle content.

The results for PC alone are also shown as dashed lines in the same figure. It is noteworthy that 10 wt% of PS blended with PC increased the expansion stress by approximately 1.4 times. The same value tended to decrease with the slight dispersion of inorganic particles into this system. This trend indicates that the decrease in the elongation at break caused by the dispersion of inorganic particles is attributable to a decrease in the breaking stress of the molded product. The inorganic particles dispersed in the PC and PS used in this study can be considered almost rigid. The particles themselves deformed only slightly during the plastic deformation, even just before rupture. This finding suggests that the inorganic particles were a source of stress concentration; they accelerated the rupture of the molded product. However, the results of the present study show that a 10 wt% blend of PS increased the fracture stress more than the fracture-promoting effect of the dispersion of the inorganic particles. Therefore, a higher fracture stress was obtained than that for PC alone.

Figure 9 shows the results obtained for the flexural strength and modulus. Blending 10 wt% PS with PC increased the flexural strength and flexural modulus. The flexural strength tended to increase when a small amount of clay was dispersed as inorganic particles. The same value decreased slightly when a small amount of silica particles was dispersed. However, the flexural modulus did not change greatly with the micro-dispersion of the clay, but it showed a tendency to decrease with the micro-dispersion of silica particles. This finding suggests that the flexural modulus tends to decrease with the fine dispersion of PS and that the clay itself has a reinforcing effect on the modulus. Table 4 shows the *E* and *υ* values of PC and PS, as estimated using the method shown in Section 2.6. The table shows that the *E* of PS was larger than that of PC, while the *υ* of PC was larger than that of PS. These results suggest that the *E* is increased by blending PS with PC and the *υ* is decreased. Table 5 presents the results of the elastic modulus evaluation of the PC/PS polymer blends. The table shows that blending PS with PC increased *E* and decreased *υ*. This trend became more pronounced when inorganic particles were added to the system.

### 3.5. Notched Impact Strength of PC/PS Polymer Blends

Figure 10 presents the effects of adding inorganic particles on the Charpy impact strength. The same figure also presents the results for PC and PS alone for comparison. The Charpy impact strength of PC blended with PS decreased but that of PS blended with a small number of inorganic particles tended to increase. In particular, when the inorganic particle content was 0.1 wt% or more, the Charpy impact strength was found to be equal to or greater than that of PC.

Taken together, these results indicate that blending PS with PC and adding a small number of inorganic particles can improve the viscosity, optical properties, and other mechanical properties while maintaining the same degree of impact resistance as for PC.

### 3.6. Morphology and Compatibility Effects of Inorganic Particles

The mechanical properties of polymer blends are governed by the morphology of the dispersed phase. For example, in the case of a polymer blend forming a sea–island structure, a coarse dispersed phase tends to cause interfacial delamination and embrittlement. In contrast, if the dispersed phase is finely dispersed, then the material can be regarded as uniformly compacted in terms of the physical properties. This induces shear yielding, which results in ductility. This is the main mechanism of compatibilization.

Inorganic particles of two types were examined for this study: plate-like clay and spherical silica. The results show that particles of both types acted as PC/PS compatibilizers. However, the amount of silica required for compatibilization was lower, indicating that spherical inorganic particles are more effective for the fine dispersion of the dispersed phase, presumably because of the interfacial tension at the particle–PC interface.

Figure 11 depicts models of the surface tension of plate-like and spherical particles. The surface tension of a plate-like particle acts parallel to the plane direction, leading to anisotropy in the surface tension of the platelets. In other words, the effect of the dispersed phase for reducing the interfacial tension depends on the platelet direction of orientation. In contrast, the surface tension of spherical particles is isotropic. Therefore, silica particles are more isotropic in reducing the interfacial tension of the dispersed phase. This point must be considered when stabilizing the quality of polymer blend molded products. Silica particles are better suited to the purposes of this study. In other words, even if the interaction force is weak, the isotropic force is more dominant and the silica contributes to the fine dispersion of PS in smaller amounts.

## 4. Conclusions

This study was conducted using inorganic particles to achieve the compatibilization of polycarbonate-based polymer blends with polystyrene as the dispersing phase. The following results were obtained.

Blending polystyrene with polycarbonate decreased the Charpy impact strength but improved the flexural strength and flexural modulus. By dispersing a small amount of inorganic fine particles such as clay or silica in the polycarbonate, the Charpy impact strength was improved to the same level as that of the polycarbonate while maintaining the transparency. Apparently, the inorganic particles promote the fine dispersion of PS. The spherical morphology of the inorganic particles was found to be more effective at compatibilizing the polymer blend because the inorganic particles apply isotropic interaction forces to the material.

## Figures and Tables

**Figure 1 materials-16-01536-f001:**
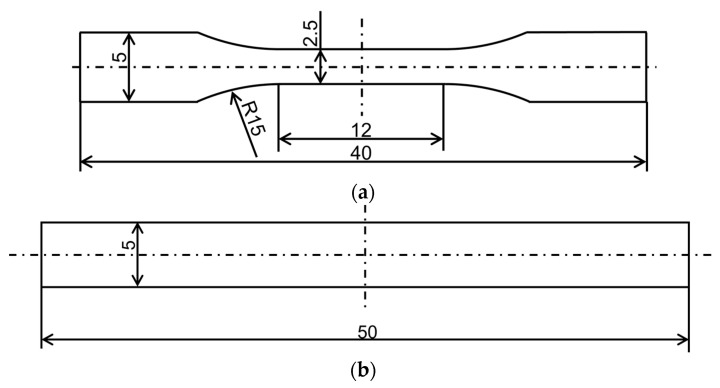
Specimen geometries: (**a**) dumbbell thickness: 1 mm; (**b**) beam thickness: 2 mm.

**Figure 2 materials-16-01536-f002:**
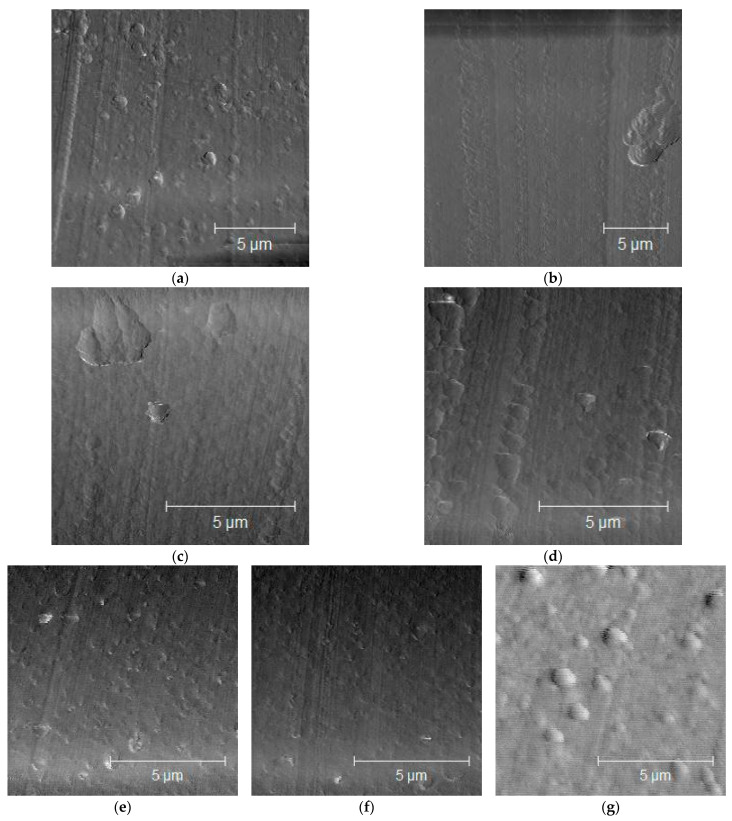
Morphologies of PC/PS/nano-fillers: (**a**) PC/PS; (**b**) PC/PS/clay (90/9.95/0.05); (**c**) PC/PS/clay (90/9.9/0.1); (**d**) PC/PS/clay (90/9.7/0.3); (**e**) PC/PS/silica (90/9.95/0.05); (**f**) PC/PS/silica (90/9.9/0.1); (**g**) PC/PS/silica (90/9.7.0.3).

**Figure 3 materials-16-01536-f003:**
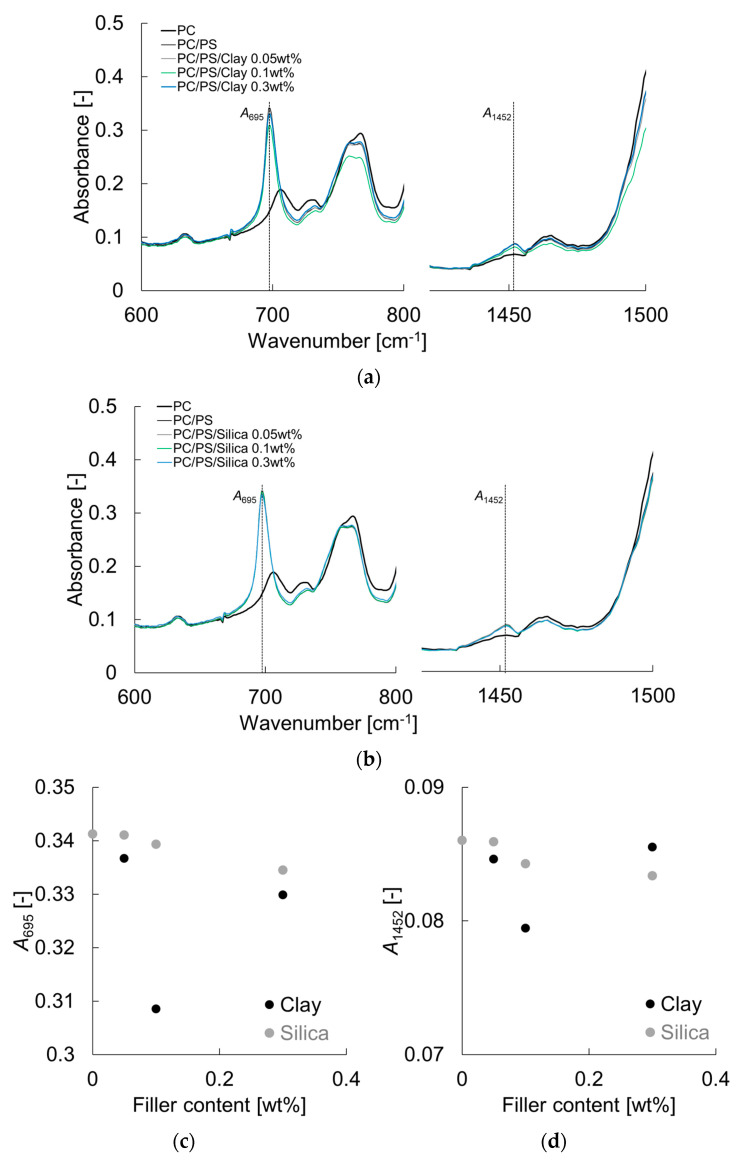
FT-IR spectra of PC/PS/clay and PC/PS/silica: (**a**) PC/PS/clay; (**b**) PC/PS/silica; (**c**) *A*_695_; (**d**) *A*_1452._

**Figure 4 materials-16-01536-f004:**
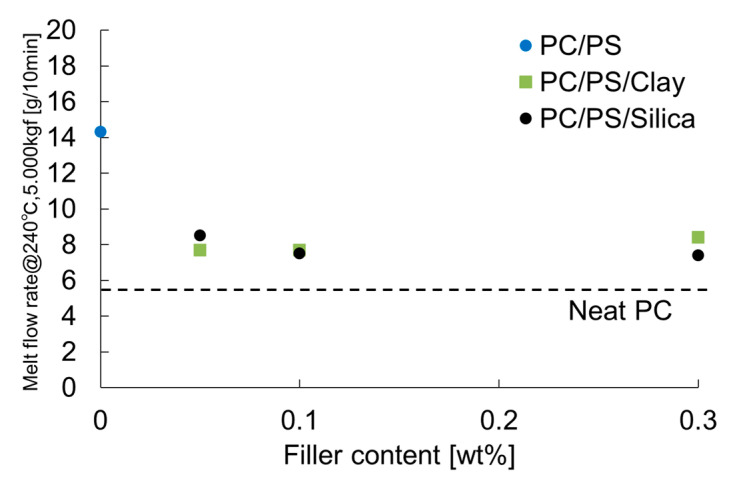
Melt flow rates of PC/PS/nano-filler composites.

**Figure 5 materials-16-01536-f005:**
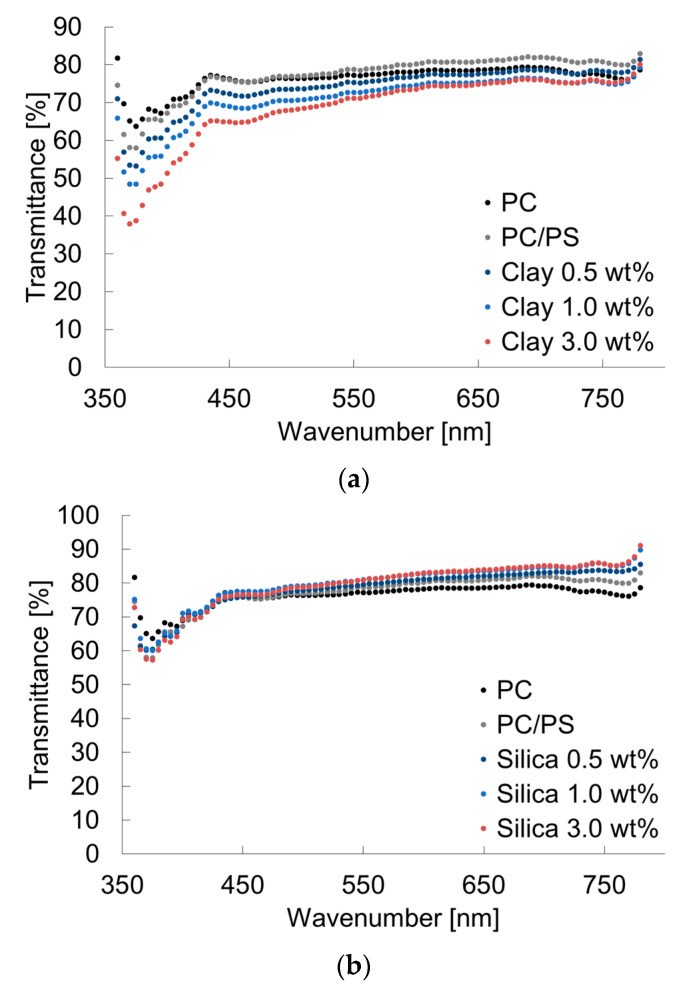
Transmittances of PC/PS/nano-filler composites: (**a**) PC/PS/clay; (**b**) PC/PS/silica.

**Figure 6 materials-16-01536-f006:**
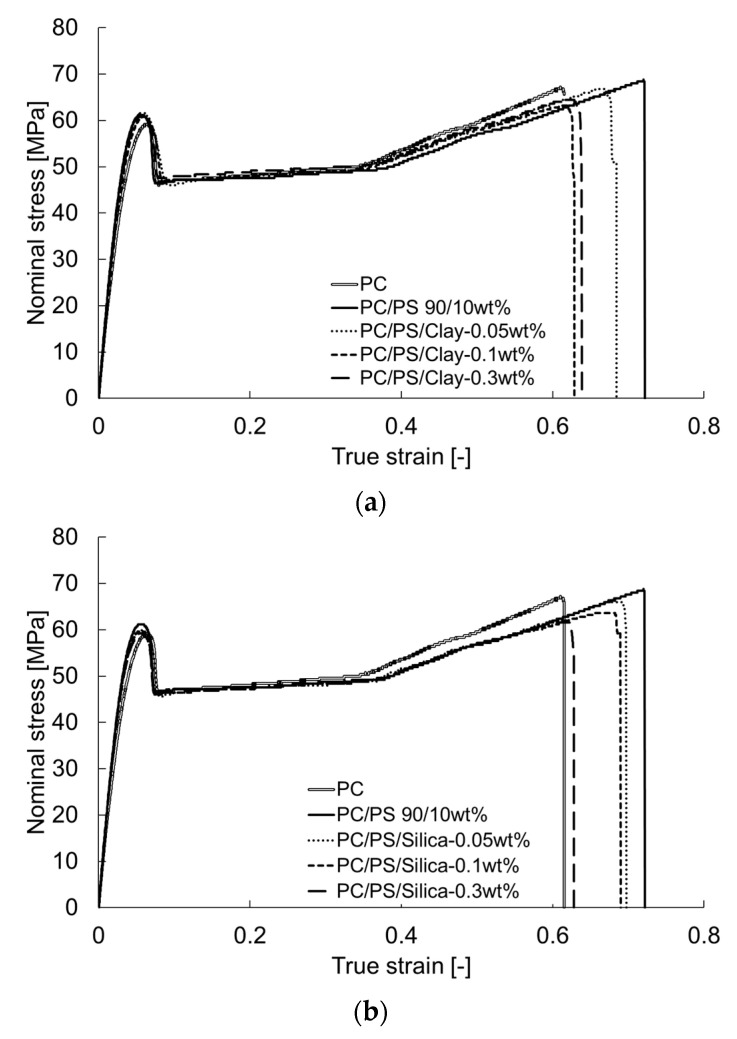
Nominal stress–true strain curves of PC/PS/nano-filler composites: (**a**) PC/PS/clay; (**b**) PC/PS/silica.

**Figure 7 materials-16-01536-f007:**
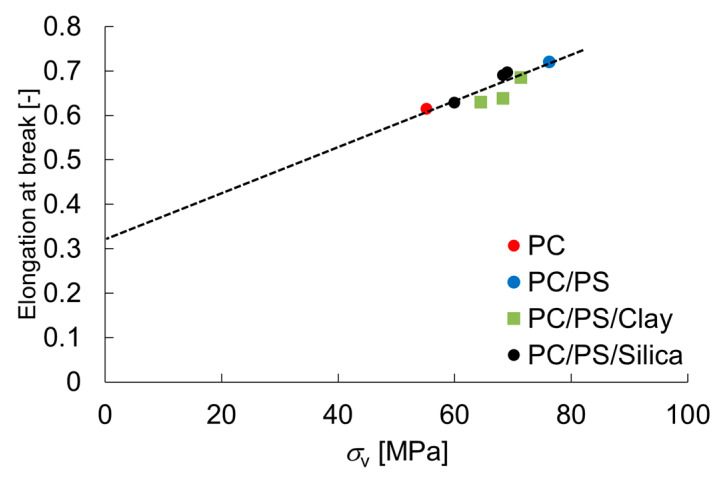
Relation between the elongation at break and stress of expansion.

**Figure 8 materials-16-01536-f008:**
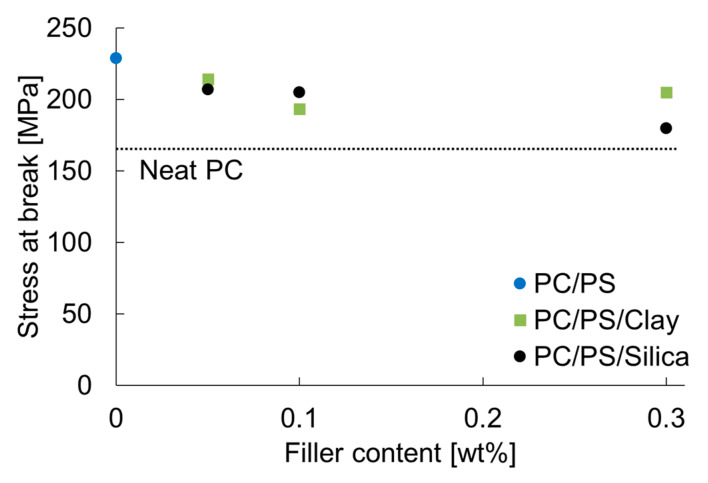
Nano-filler content dependence of the stress at break.

**Figure 9 materials-16-01536-f009:**
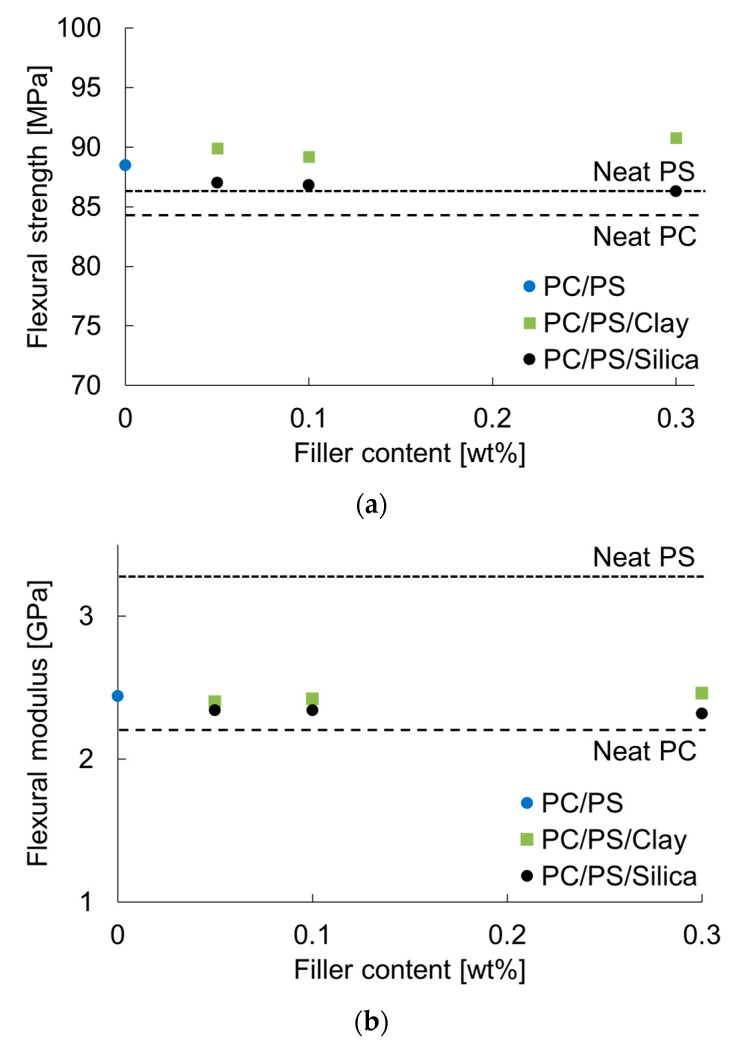
Nano-filler content dependence of flexural properties: (**a**) flexural strength; (**b**) flexural modulus.

**Figure 10 materials-16-01536-f010:**
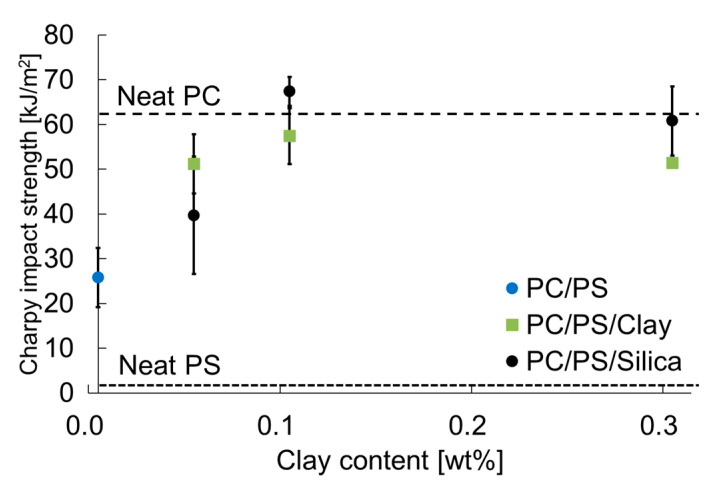
Nano-filler content dependence of the Charpy impact strength.

**Figure 11 materials-16-01536-f011:**
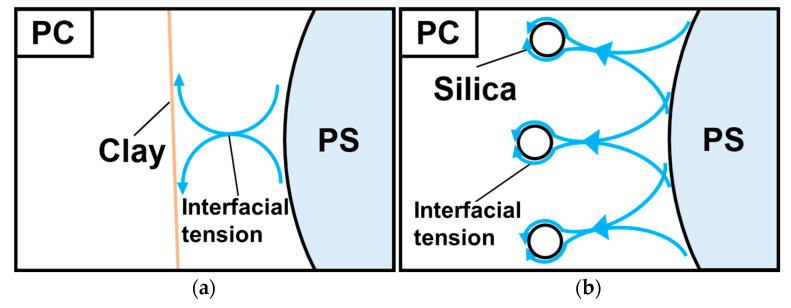
Models of the interfacial tension of plate-like and spherical particles: (**a**) PC/PS/clay; (**b**) PC/PS/silica.

**Table 1 materials-16-01536-t001:** Material compositions.

PC [wt%]	PS [wt%]	Clay [wt%]	Silica [wt%]
100	-	-	-
90	10	-	-
90	9.95	0.05	-
90	9.9	0.1	-
90	9.7	0.3	-
90	9.95	-	0.05
90	9.9	-	0.1
90	9.7	-	0.3

**Table 2 materials-16-01536-t002:** Injection molding conditions.

Parameter	Dumbbell	Beam
Injection temp. [°C]	310	310
Mold temp. [°C]	90	110
Injection speed [mm/s]	10	30
Holding pressure [MPa]	105	105
Injection time [s]	15	15
Cooling time [s]	10	10

**Table 3 materials-16-01536-t003:** Glass transition temperatures of PC (*T*_g,PC_) and PS (*T*_g,PS_).

	*T*_g,PS_ [°C]	*T*_g,PC_ [°C]
PC	-	152.4
PC/PS	106.5	149.5
PC/PS/Clay 0.1 wt%	102.2	150.6
PC/PS/Silica 0.1 wt%	104.1	150.7
PS	98.3	-

**Table 4 materials-16-01536-t004:** Elastic properties of PC and PS.

Materials	*T*_inj_ [°C]	*FS*^※^ [MPa]	*FM*^※^ [MPa]	*E* [MPa]	*υ* [-]
PC	310	84.4	2250	1260	0.373
PS	310	86.7	3330	1570	0.399

^※^ FS and FM mean the flexural strength and flexural modulus, respectively.

**Table 5 materials-16-01536-t005:** Elastic properties of PC/PS/clay and PC/PS/silica composites.

PC [wt%]	PS [wt%]	Clay [wt%]	Silica [wt%]	*T*_inj_ [°C]	*FS* [MPa]	*FM* [MPa]	*E* [MPa]	*υ* [-]
90	10	-	-	310	88.5	2440	1380	0.371
90	9.95	0.05	-	310	89.9	2400	1380	0.367
90	9.9	0.1	-	310	89.2	2420	1380	0.369
90	9.7	0.3	-	310	90.8	2460	1420	0.368
90	9.95	-	0.05	310	87	2340	1320	0.371
90	9.9	-	0.1	310	86.8	2340	1320	0.371
90	9.7	-	0.3	310	86.3	2320	1310	0.371

## Data Availability

The data presented in this study are available on request from the corresponding author.

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
