# Peer review of "Inorganic Particles Contribute to the Compatibility of Polycarbonate/Polystyrene Polymer Blends"

_materials, 2023, doi:10.3390/ma16041536_

Round 1

Reviewer 1 Report

The manuscript could interest the readership of this journal, if the authors are ready to consider the following suggestions:

1. Minor typos, phrase and punctuation errors should be corrected throughout the manuscript.

2. Authors should avoid repeating parts of title words as keywords.

3. Novelty of this study must be highlighted under Introduction section.

4. The manuscript must be clearly sectionalised into Abstract, Introduction, Materials and methods, Results and discussion, Conclusions and References

6. Arabic numerals for the sections and subsections must be corrected throughout the manuscript.

Author Response

Thank you for reviewing this paper. I would like to respond to the points you raised in your review. I have also made additional corrections to the points raised by other reviewers, and would appreciate your reviewing them as well.

  1. Minor typos, phrase and punctuation errors should be corrected throughout the manuscript.

> Thank you for pointing this out. I have corrected the typographical and punctuation errors with the utmost care.

  1. Authors should avoid repeating parts of title words as keywords.

> Thank you for pointing this out. I have updated the keywords so that they do not overlap with the title as much as possible.

  1. Novelty of this study must be highlighted under Introduction section.

> Thank you for your suggestion. I have added a note in the "Introduction" section regarding the ingenuity of the molding process used for the microdispersion of inorganic particles proposed in this article.

  1. The manuscript must be clearly sectionalised into Abstract, Introduction, Materials and methods, Results and discussion, Conclusions and References

> Thank you for pointing this out. I have reorganized the text so that it is as you indicated.

  1. Arabic numerals for the sections and subsections must be corrected throughout the manuscript.

> Thank you for pointing this out. I have modified the sections and subsections so that they are as you indicated.

Reviewer 2 Report

Dear Editor, in the submitted paper PC/PS blends have been prepared and studied. To achieve compatibility inorganic particles like clay and silica have been added. From the results it is clear that the inorganic particles promote the fine dispersion of PS and thus mechanical properties have been enhanced. The paper is well organized and provides some new and interesting data. For this reason I propose to accept it for publication.

It is not clear why only the blend ratio PC/PS 90/10 w/w was studied.

Both polymers are amorphous. However, after the addition of nanoparticles did the authors observe any effect on their glass transition values?

I think that FTIR spectra should be added to prove the claimed interactions in the interface.

Author Response

Thank you for reviewing this paper. I would like to respond to the points you raised in your review. I have also made additional corrections to the points raised by other reviewers, and would appreciate your reviewing them as well.

  1. It is not clear why only the blend ratio PC/PS 90/10 w/w was studied.

> Thank you for pointing this out. The reason why I only studied PC/PS 90/10 w/w in this paper is that previous studies have reported that a 10 wt% PS blend can provide sufficient viscosity reduction to improve moldability. Therefore, this reason was added to the experimental methodology.

  1. Both polymers are amorphous. However, after the addition of nanoparticles did the authors observe any effect on their glass transition values?

> Thank you for pointing this out, I did additional DSC and investigated the change in Tg due to inorganic particle dispersion for PC and PS. As a result, I obtained a tendency for the Tg of PS to decrease and the Tg of PC to increase.

  1. I think that FTIR spectra should be added to prove the claimed interactions in the interface.

> Thank you for pointing this out, I did an additional FTIR and examined the change in the two absorption peaks derived from PS. As a result, each peak became smaller with the addition of a small amount of inorganic particles, and the trend was more pronounced for clay. The Tg of PS was significantly lower in clay, where a significant peak reduction occurred. From these two trends, I considered that the clay particles have a larger interaction force. The reason for this is also discussed in terms of particle size and added to the text.

Reviewer 3 Report

In this manuscript, authors demonstrated the contribution of inorganic particles in compatibility of PC/PS polymer blends, which could an useful method to improve the PC/PS polymer. However, some questions needed to be solved before publishing.

Question 1: the front and size of words in table should be consistent to the maintext. 

Question 2: the numbering of your sections and headings are wrong, which needs to be revised.

Question 3: do you use any methods to confirm the compositions of the prepared materials? I wonder if your clay or silica ratio is accurate. 

Question 4: the figure 1g is not clear, which should be updated to high resolution image.

Question 5: in figures 3,6,10, the black dot of PC/PS/Silica at O wt% filler content are missing. Maybe it is overlapped with PC/PS/Clay? If so, you should note in figure capture.

The manuscript needs to be revised, including update language and correct typos.

Author Response

Thank you for reviewing this paper. I would like to respond to the points you raised in your review. I have also made additional corrections to the points raised by other reviewers, and would appreciate your reviewing them as well.

Question 1: the front and size of words in table should be consistent to the maintext.

> Thank you for pointing this out. The size of the figure was adjusted so that it would be appropriately sized.

Question 2: the numbering of your sections and headings are wrong, which needs to be revised.

> Thank you for pointing this out. I have modified the sections and subsections so that they are as you indicated.

Question 3: do you use any methods to confirm the compositions of the prepared materials? I wonder if your clay or silica ratio is accurate.

> Thank you for pointing this out. Since the amount of inorganic particles added in this paper is extremely small, less than 1 wt%, I believe that it is difficult to precisely adjust the composition by dry blending. Therefore, I once prepared a master batch with PS/filler = 90/10 w/w and mixed the master batch with pellets of PC and PS to achieve the composition shown in Table 1. Although I have not directly confirmed whether the composition is as shown in Table 1, I believe that it is reasonably well adjusted based on the mechanical properties and the additional FT-IR results I measured.

Question 4: the figure 1g is not clear, which should be updated to high resolution image.

> Thank you for your suggestion. I have improved the resolution of Figure 1.

Question 5: in figures 3,6,10, the black dot of PC/PS/Silica at O wt% filler content are missing. Maybe it is overlapped with PC/PS/Clay? If so, you should note in figure capture.

> Thank you for pointing this out.  I have indicated the 0wt% data with a different symbol as "PC/PS".

The manuscript needs to be revised, including update language and correct typos.

> Thank you for pointing this out. I have corrected the typographical and punctuation errors with the utmost care.

Round 2

Reviewer 1 Report

Accept

Author Response

Thank you for your review. I am grateful for the decision to accept the manuscript. I have added the equipment details pointed out by the Academic Editor. The layout has been changed to reflect the addition.
